# Key Invariants in the Evolution of Sociality Across Taxa

**DOI:** 10.3390/biology14091239

**Published:** 2025-09-10

**Authors:** Bianca Bonato, Marco Dadda, Umberto Castiello

**Affiliations:** Department of General Psychology, University of Padova, Via Venezia 8, 35131 Padova, Italy; marco.dadda@unipd.it (M.D.); umberto.castiello@unipd.it (U.C.)

**Keywords:** sociality, competition, cooperation, social behavior, aneural organisms, plant behavior, social modes evolution

## Abstract

Social behavior has evolved in many forms across the living world. This review explores what different types of life, including bacteria, single-celled organisms, animals, and plants, can tell us about how social traits emerge. By comparing examples across major groups of life, we show that similar challenges in the environment may lead to similar social strategies, even in species that are not closely related. Understanding these shared features can help scientists uncover general principles about how social life begins and is maintained in nature.

## 1. Introduction

A central objective in biology is to elucidate the evolutionary processes underlying complex traits, such as the emergence of social behaviors [1]. Contemporary research on the evolution of sociality seeks to integrate species-specific findings into broader, more generalized theoretical frameworks. However, formulating a unified theory of social evolution remains a formidable challenge due to the intrinsic complexity of social systems. Here, we begin this endeavor by adopting a definition of social behavior as any action of an individual that influences the behavior of other members of the same species, whether through cooperative or competitive interactions [2] (see Appendix A).

One promising approach conceptualizes social organization as the cumulative outcome of individual decisions—such as whether to disperse from the natal territory, to engage in cooperative breeding, or to provide alloparental care. This decision-making framework, or “social trajectory,” facilitates cross-taxon comparisons by allowing researchers to examine whether similar selective pressures influence key social decisions across diverse lineages [2].

To further unravel the mechanisms underlying social behavior, neuroscience has increasingly focused on the brain’s role in mediating social cognition. In a seminal review, Brothers and colleagues [3] proposed the existence of a specialized network of brain regions, the “social brain”, comprising the amygdala, the orbitofrontal cortex, and the temporal cortex. This proposal was initially supported by studies in non-human primates (i.e., *Macaca Rehsus*; *Macaca Speciosa*). For example, lesions to the amygdala result in social withdrawal [4], while damage to the orbitofrontal cortex alters social behavior [5]. Moreover, in non-human primates the mirror neurons within the ventral premotor cortex, permitting the representations of others’ actions [6,7] and neurons in the superior temporal sulcus that are selectively responsive to facial expressions and gaze direction [8] have been discovered.

With the advent of neuroimaging, these findings have been extended to human studies, reinforcing the validity of the hypothesis of Brothers et al. [3,9]. A major addition to the social brain network has since emerged, the medial prefrontal cortex and the paracingulate cortex—both implicated in theory of mind tasks [10]. The concept of a “social brain” specialized for navigating social environments is thus strongly supported by converging evidence from neuroimaging [4,5] and, to a lesser extent, lesion studies [4,5,9].

While it is often assumed that the evolution of sociality is tightly linked to the emergence of advanced cognitive capacities and complex brain structures, growing evidence challenges this notion by demonstrating that intricate social behaviors can arise from relatively simple neural architectures. For instance, invertebrate species such as social insects—ants, bees, and termites—exhibit sophisticated forms of social organization including cooperative brood care, division of labor, and collective decision-making, despite possessing miniature and non-centralized nervous systems [11]. These findings suggest that social coordination does not necessarily require high-level cognitive processing, but can instead be governed by decentralized neural circuits and simple behavioral rules.

This principle extends to vertebrates as well. To date, complex social dynamics such as schooling in fish [12,13] and flocking in birds [14,15] emerge from local sensorimotor interactions and distributed control mechanisms rather than from top-down executive functions. These behaviors illustrate how structured and coordinated group activity can arise from bottom-up processes embedded in relatively rudimentary neural systems [11,12,13].

Remarkably, even in humans, arguably considered the species with the most complex brain, social behavior appears to manifest at an unexpectedly early stage of development. Castiello and colleagues [16] documented that during twin pregnancies, as early as the 11th week of gestation, fetuses engage in purposeful, goal-directed movements toward their co-twin. These interactions show both spatial and temporal coordination, implying that the fetus is capable of rudimentary social motor planning well before the maturation of the neocortex and other higher-order brain structures. This observation strongly indicates that the roots of human sociality are embedded in fundamental sensorimotor capacities and that social engagement may be a deeply embodied and developmentally precocious phenomenon.

Taken together, these findings challenge the assumption that sociality arises exclusively from advanced neural computation. Instead, they support the view that the capacity for social interaction can be scaffolded by relatively basic neural systems, and in humans, emerges far earlier than previously thought—prior to birth and before the full maturation of complex brain circuits. Such evidence points toward a more foundational, perhaps evolutionarily conserved, basis for social behavior, grounded in early-developing mechanisms of perception, movement, and interaction (Figure 1).

In light of the evidence that social behavior can arise independently of complex neural systems—even in species with advanced brains such as humans—it becomes both relevant and necessary to broaden the scope of inquiry to include organisms that entirely lack a nervous system. In this context, we turn our attention to aneural organisms such as bacteria and plants, whose social capacities, offer crucial insights into the evolutionary underpinnings of sociality. This line of investigation is justified by the presence of two fundamental prerequisites for social interaction in these life forms: the ability to distinguish self from non-self, interpreted as an active form of proprioceptive elaboration, and the capacity to discriminate between kin and non-kin [17,18,19,20,21,22,23]. These features, which are essential for modulating interaction strategies such as cooperation or competition, form the core of social engagement even in the absence of a centralized nervous system.

## 2. Self/Non-Self Recognition

Because social behavior can be defined as the action of one individual that influences the behavior of another [2], its very existence depends on the ability to distinguish self from non-self. This recognition is a prerequisite for directing cooperation toward appropriate partners, maintaining group integrity, and defending against exploiters or pathogens. Comparative evidence across taxa reveals that, although proximate mechanisms vary, a common theme underlies self/non-self-recognition: the mediation of interactions through molecular or chemical cues that are evaluated against an internal reference template.

All recognition systems must operate with precision [24], and such precision is typically maintained through extensive polymorphism in the phenotypic traits used for recognition [25]. When individuals encounter one another, several components interact to produce the appropriate response [24,25,26]. The evaluator (actor) assesses a phenotypic label expressed by another individual (the receiver). Labels may include chemical odors, cell-surface proteins, acoustic signals, color patterns, or stereotyped displays. The evaluator compares these labels against a self-referential template: a match results in acceptance [27], whereas a mismatch triggers rejection. Different recognition systems may apply different matching rules, with varying tolerance to partial similarity. In social contexts, shared labels often reflect kinship, and self/non-self-recognition thereby becomes kin recognition.

In the microbial world, compelling examples highlight how recognition systems underpin social behavior. The bacterium *Proteus mirabilis*, for instance, exhibits strain-specific territoriality during swarming. When genetically distinct colonies meet, they form stable demarcation boundaries—Dienes lines—that demarcate self from non-self [25]. These sharp boundaries are actively maintained by inter-strain antagonism, reflecting a rudimentary form of conflict resolution grounded in molecular discrimination. Similarly, kin discrimination proteins in *Bacillus subtilis* or outer-membrane receptors in *Myxococcus xanthus* restrict cooperative behaviors, such as biofilm formation or fruiting body development, to closely related cells, thereby preventing exploitation by non-kin lineages [25].

Plants also rely heavily on molecular recognition. At the reproductive level, self-incompatibility (SI) systems prevent self-fertilization by rejecting pollen carrying matching alleles, thereby promoting outcrossing and maintaining genetic diversity as demonstrated in *Brassicaceae* family (*B. rapa*, *B. oleracea*; [28]). At the level of resource competition, plants can distinguish self from non-self roots, modulating growth accordingly—suppressing competition with self while intensifying responses to non-self neighbors as in *Cakile Edentula* [29]. Their immune systems add another layer: innate receptors such as PRRs (Pattern Recognition Receptors) and NLRs (Nucleotide-binding Leucine-rich repeat Receptors) recognize pathogen-associated molecular patterns, triggering defense, yet also permit recognition of beneficial symbionts in *Arabidopsis Thaliana* [30]. Notably, SI systems in plants show parallels to other recognition systems—such as Hymenopteran sex determination [31,32] and fungal mating systems [33]—all characterized by strong selection maintaining extensive allelic polymorphism [25].

In social insects, recognition systems achieve even greater complexity but remain fundamentally chemical. In ants (e.g., *Camponotus vagus*, *Camponotus floridanus*), bees (e.g., *Apis Mellifera*; *Lasioglossum zephyrum*) and wasps (e.g., *Polistes* spp., *Vespuls* spp.), cuticular hydrocarbons (CHCs) constitute colony-specific odor signatures, providing the primary labels for nestmate recognition [25,26]. These chemical profiles are learned and constantly updated, allowing workers to discriminate nestmates from intruders and maintain social cohesion. Variation in CHC blends can encode not only colony membership but also caste identity and even task-specific roles [34,35]. Such finely tuned chemical recognition systems illustrate the sophistication of molecular communication in eusocial contexts.

Across these diverse examples—from bacteria to plants to insects—a unifying principle emerges: self/non-self-recognition is predominantly based on chemical or molecular cues, which act as phenotypic labels compared against an internal template. Whether through polymorphic proteins in bacteria, receptor–ligand incompatibility systems in plants, or cuticular hydrocarbon blends in insects, these mechanisms provide the foundation for cooperation, competition, and the evolution of social complexity, demonstrating that social behavior across the tree of life is rooted not only in neural computation but also in deeply conserved molecular recognition strategies (see Table 1).

## 3. Communication and Signaling

The evolution of communication and signaling mechanisms plays a fundamental role in the emergence and maintenance of social behaviors. Effective social interactions, whether cooperative or competitive, depend on the accurate transmission and reception of information between individuals. Traditionally, studies on the evolution of signaling have emphasized the properties of the signal itself, exploring how it conveys information about the signaler’s attributes [36]. However, communication is inherently a two-way process involving both signalers and receivers, and theoretical models predict co-evolutionary dynamics between them. Features of the receiver impose selective pressures on the design of the signal, while the signal, in turn, selects for the receiver’s ability to detect and interpret it. This interplay is often reflected in the structure and sensitivity of receptors or receptor organs [37].

Communication can be chemical, visual, or physical among organisms, and it helps to coordinate activities, designate group membership, or identify individuals or their roles in a group [38].

### 3.1. Communication and Signaling in Microbial Domain

In microbial domain, Bacteria can produce an extensive repertoire of secondary metabolites, and can respond to a wide variety of chemicals in their environment. In recent years, particular groups of secondary metabolites have been characterized for their role in the regulation of gene expression in a cell-density-dependent manner, and this behavior has been collectively referred to as quorum sensing (QS), or cells communication strategy.

Quorum sensing is generally thought to act as a mechanism for the coordinated regulation of behavior at the level of populations of cells, enabling bacteria to communicate, cooperate and alter their behavior and the one of others, according to changes in their social environment [36]. Currently, there are three well-defined classes of molecules that serve as the paradigms for chemical signaling in bacteria: oligopeptides, AHLs (Acyl-Homoserine Lactones) and the LuxS/autoinducer-2 (AI-2) class [36]. It is important to realize that the mere demonstration of bacteria responding to a chemical substance produced by other bacteria does not necessarily imply communication. It is important to differentiate between signals, cues and chemical manipulation, which differentially affect the fitness of the emitter and the receiver. 

Specifically, in Bacteria, communication seems to occur when there are intraspecific interactions between bacteria. An experimental study with the pathogenic bacterium *P. aeruginosa* showed that higher relatedness results in higher levels of cooperation, as demonstrated by a higher investment in the production of siderophore iron-scavenging agents in lines that are kept under higher relatedness [39]. Similarly, experiments in which strains of *Myxococcus Xanthus* competed against one another in all possible pair-wise combinations showed that in most pairings, at least one competitor showed strong antagonism towards its partner [40]. These data show that bacteria can perceive the presence of different strains and that changes in overall relatedness might have profound effects on population growth and survival.

Similarly, bacteria in the environment might use quorum sensing to regulate expression of extracellular enzymes that degrade macromolecules. As both the enzymes and their products will diffuse away from the cell or cells, there is an advantage to initiating catabolic production when there are sufficient numbers of cells to scavenge all the products [36]. This is clearly more efficient in a microcolony than in single isolated cells, and coordinating the expression of these enzymes using intraspecies signaling systems would be optimal.

There are examples, however, of signaling within groups of non-clonal bacteria, such as *M. xanthus*. In these organisms, individual cells aggregate under starvation conditions and form fruiting bodies. Within these fruiting bodies, some cells develop into spores and the others die [41,42,43]. In *M. xanthus*, this process is mediated by two signaling pathways: the first (A-signaling) leads to aggregation of cells, and the second (C-signaling) involves the formation of a mound and, ultimately, fruiting-body formation [42,44]. Cheaters that are defective in either A- or C-signaling have been identified which are overrepresented in the spore-forming population when grown in mixed colonies with wild-type cells. When grown in monoculture, they produce spores at a lower frequency than wild-type cells [45]. If one considers the wild-type behavior to be altruistic, then these cheaters take advantage of wild-type cells, but fare less well on their own.

### 3.2. Communication and Signaling in Plants

Remaining within the domain of aneural organisms, recent evidence further supports the notion that communication is not necessarily contingent upon the presence of a brain. Notably, plants have also been shown to engage in complex forms of communication and signaling. These interactions occur both above and below ground through various mechanisms, including the release of chemical signals, root system interactions, and the use of mycorrhizal networks [46,47,48,49,50].

The most efficient and complex mechanism plants use to communicate is via VOCs, a chemical signaling that conveys multiple meanings [48]. Plants (e.g., *Phaseolus vulgaris*, *Zea mays*, *Nicotiana tabacum*, *Oryza sativa*, *Pisum Sativum*, *Gossypium hirstum*) can communicate using VOCs for a variety of reasons, including responding to predator attacks [51,52,53,54], attracting pollinators [54], exchanging useful information [55], and adapting to environmental stress [56]. Although these subtle chemical signals are more effective among genetically related individuals [57], unrelated or “stranger” plants also appear to eavesdrop on these cues and use the information to initiate adaptive responses that enhance their own survival [58,59,60]

Terpenoids, the most diversified of these families of chemicals, have an integral number of 5-carbon units that are shared by all plants and are engaged in internal and external communication as well as plant defense [21]. Terpenoids are released in reaction to internal and external variables, and the information or effects of these terpenoids are sensed by other portions of the plant, as well as other plants, animals, and/or microbes [51]. Some plant terpenoids are key chemical agents in plant communication because they play a function in mediating a variety of ecological interactions [51]. Importantly, in terms of ecology and biology, the concentration of terpenoids changes depending on the sort of message to be delivered. Terpenoids, for example, attract insects only when low amounts are released and become more repellent to pollinators at increasing concentrations [61].

Plants may synthesize a wide range of volatile chemical compounds by combining these fundamental 5-carbon units in various combinations [62], for example, highlighted the fact that plant language has real combinatorial flexibility, which means that new meanings may be attributed to old chemical terms and employed in new settings, resulting in fresh interactions. Inducible VOCs are employed in plant-to-plant signaling, pathogen defense, and ozone quenching, in addition to attracting natural enemies of herbivores, as well as tropospheric ozone and fine-particle aerosol formation. In evolutionary terms, the inventory and the various combinations of chemical utterances were enhanced with meaning and passed down through generations exactly via usage and experience in a range of settings [48].

Further, synomons are allelochemicals that are adaptable to both the transmitter and the receiver [51]. Herbivore-induced plant volatiles are scents generated by attacked plants that serve as key indications for herbivorous insect predators to identify their prey [63]. Terpenoids are found in several volatile synomones [52,64]. Each plant species and cultivar develop its own distinct mix of synomones caused by herbivores, which implies that predators encounter a variety of synomones depending on the host’s diversity [65]. Fascinating research assessing population-specific VOC emissions was recently devised and carried out [59]. Based on the assumption that plants respond to volatile cues emitted by damaged neighbors in order to increase their defense against herbivores, some researchers attempted to determine whether plant communication is more effective with local versus distant neighbors [59]. For example, it has been discovered that sagebrush tissues (*Artemisia tridentata*) increased their resistance to herbivory by responding to volatile signals emitted by experimentally wounded neighboring plants [59,66]

Below ground, chemical signaling is equally critical. Recent research has identified strigolactones (SLs), an endogenous plant hormone that regulate various aspects of plant growth and development [67,68,69], as key signaling molecules in root communication. Plants can detect SLs concentrations in the soil, allowing them to assess the presence and identity of neighboring plants and adjust their growth accordingly [70]. SLs also mediate internal signaling from roots to shoots [71], and their abundance may function as a cue of local competition for below-ground resources [72].

Despite belonging to distinct domains of life, bacteria employing quorum sensing (QS) and plants releasing volatile organic compounds (VOCs) exhibit strikingly convergent strategies of chemical communication. Both systems rely on the production, release, and detection of diffusible small molecules that mediate population- or community-level coordination. 

It is worth noticing that several mechanistic parallels underscore the functional analogy between QS and plant VOC signaling. (i) Concentration dependence and threshold responses: both systems translate extracellular molecular abundance into coordinated group-level outcomes, ensuring responses are activated only under ecologically relevant conditions. (ii) Information integration: QS molecules and VOCs act as multiplexed signals, providing nuanced information beyond simple presence/absence. (iii) Context sensitivity: responses are not uniform but shaped by the developmental stage, genotype, or prior experience of the receiver, enabling fine-tuned behavioral plasticity.

However, it is necessary to say that at present, no studies have reconstructed an evolutionary pathway linking bacterial QS to plant volatile-mediated interference.

In our opinion, it would be interesting to investigate whether, in different taxa, such mechanisms of chemical competition and communication reflect shared ancestry, parallel co-option of similar biochemical pathways, or true convergence leading to similar ecological outcomes despite distinct mechanistic origins [73].

### 3.3. Communication and Signaling in Animals

Communication in *Animalia* kingdom presents several techniques and signal employed. In animals, communication integrates multiple modalities, including chemical, visual, auditory, and vibrational signals. In social insects such as ants (e.g., *Camponotus vagus*, *Camponotus floridanus*, *Linepithema humile*, *Solenopsis invicta*), bees (e.g., *Apis mellifera*, *Lasioglossum zephyrum*), and wasps (e.g., *Polistes* spp., *Vespula* spp.), pheromones and chemical odors, particularly cuticular hydrocarbons, serve as colony-specific labels that enable nestmate recognition, task allocation, and maintenance of social hierarchy [25,26,34]. These chemical signals ensure the cohesion of complex eusocial systems. Pheromones is not only a matter of insects. Indeed, they are found across all forms of life, from bacteria to mammals, underscoring their fundamental role in intercellular communication. In fungi (*Saccharomyces cerevisiae)*, they are essential for coordinating interactions between mating partners and guiding the process of sexual reproduction [74].

In fish (e.g., *Ostariophysi*, *Cichlidae*, *Gobidae*, *Pomacentridae*, *Porichthys notatus*), chemical cues, sound and visual displays facilitate shoaling, territoriality, and mate choice [75,76]. While in birds (e.g., *Parus minor*, *Troglodytes aedon*, *Taeniopygia guttata*, *Parus* spp., *Junco hyemalis*), vocalizations, plumage visual patterns, and chemical signals mediate territory defense, reproductive coordination, and social affiliation [77,78,79].

Across these diverse taxa, the evolution of communication reflects a co-adaptive process: the features of the signal must be interpretable by the receiver, and the sensory structures of the receiver evolve to detect and respond accurately to the signal.

In our opinion, despite differences in modality and complexity, several key invariants emerge across taxa. Chemical and molecular cues form a foundational basis of communication, spanning bacterial quorum sensing, plant volatile organic compounds, and the cuticular hydrocarbons of social insects [80]. These cues are evaluated against internal templates, enabling individuals to discriminate relevant signals from background noise. Communication also universally involves selective pressures on both signaler and receiver, driving co-evolutionary dynamics. Signals exert context-dependent effects on the behavior of recipients, shaping cooperation, competition, reproduction, and social organization. At the same time, the integration of signal production, detection, and response establishes a functional architecture that supports social coordination regardless of neural complexity. Collectively, these principles underscore that chemical and molecular signaling provides the mechanistic foundation for social communication, which is subsequently expanded and diversified through visual, auditory, and multimodal channels in animals. Across bacteria, plants, insects, fish, and birds, effective communication underpins the emergence, maintenance, and evolution of social behavior (see Table 2).

## 4. Cooperation in Collective Construction & Resource Sharing

Allowed by self/non-self-recognition and communication basic skills, cooperation can emerge. Cooperative behavior lies at the core of social behavior, enabling organisms to secure resources, defend against threats, reproduce, sharing resources and colonize new environments [81,82]. This kind of cooperation can be egalitarian, with individuals contributing and benefiting equally, or structured through division of labor, where roles and rewards differ [83]. While traditionally studied in macroscopic animals, recent work has extended the study of cooperative behavior to aneural organisms such as microbes and plants, revealing sophisticated strategies [84].

### 4.1. Cooperation in Microbial Domain

Among microorganisms, a particularly striking example of cooperative behavior is found in the Protista kingdom, in the amoeba *Dictyostelium discoideum*. When faced with starvation, individual amoebae aggregate to form a multicellular structure known as a fruiting body, composed of a stalk and a mass of spores. While the spores survive and remain capable of germinating once environmental conditions improve, the stalk cells undergo programmed cell death. Although this does not represent reproduction in the conventional sense, the sacrifice of stalk cells enhances the group’s overall reproductive success by supporting the elevation and dispersal of spores [85,86].

Notably, these aggregations can consist of genetically unrelated individuals, yet cooperative behavior and functional specialization still emerge. Cells that adopt the non-reproductive stalk role generally retain the potential to reproduce under certain conditions, indicating that their behavior represents an active investment in collective success rather than simply being a consequence of incapacity or terminal state. Nonetheless, this system is prone to conflict: mixed groups are susceptible to “cheater” cells that contribute disproportionately to spores while avoiding stalk formation. These dynamics highlight the delicate balance between cooperation and competition, demonstrating how natural selection can favor division of labor even among potentially competing genotypes [45,87].

A comparable phenomenon is seen in the Bacteria domain with *Myxococcus xanthus*, a bacterium that preys on other microbes and forms complex fruiting bodies during nutrient scarcity. Within these structures, some cells become spores, others die autolytically, and some remain active as protective cells on the periphery [88]. The energetic cost of producing fruiting bodies creates opportunities for cheaters, including obligate social cheats that cannot sporulate independently but survive by joining cooperative strains. These cheaters, through novel mutations rather than reversions, can evolve into new social types, illustrating the dynamic evolution of cooperation in unicellular organisms [88].

Central to microbial cooperation is communication. In both *Dictyostelium* and *Myxococcus*, multicellular development is orchestrated by chemical signals and context-dependent physical interactions, demonstrating that complex behaviors can arise from relatively simple systems influenced by environmental cues that facilitate developmental and behavioral plasticity [44,89,90,91]. Cooperative and communication-mediated interactions have also been identified in *Archaea*. Sharma et al. [92] demonstrated that archaeal communities coordinate to reach a quorum—the starting point for group decision-making. In human oral microbiomes, cooperation and competition occur within diverse communities of ca. 500 species coordinating all together [93].

### 4.2. Cooperation in Plants

Cooperative behavior is also evident in *Eukarya*. Plants suppress their own growth in the presence of kin, reducing competition and potentially enhancing collective fitness [94,95] (Figure 2B). Such adjustments, downregulating growth near relatives, may improve reproductive success for all involved [96]. Such kin-favoring behavior is also observed in the algae *Volvox carteri*, where daughter colonies develop within the parent colony and are released upon maturation, suggesting early or incipient forms of parental investment [96,97].

In addition to kin-sensitive behaviors, recent evidence suggests that cooperative tendencies can also manifest in the movement patterns of climbing plants (Figure 2B). Bonato et al. [23], employing the 3D kinematic analysis previously used to study competitive interactions [22], investigated potential motor coordination strategies in *Pisum sativum* under controlled experimental conditions. In this study, no external support structure was provided; instead, two pea plants were placed at equal distances under balanced conditions of light, nutrients, water, and soil. The only viable strategy for vertical growth, and thus access to light, was mutual entwinement of their tendrils. This behavior, known as intertwining [98], involves the formation of a braided structure through which the two plants support each other mechanically as they grow upward. The kinematic reconstruction revealed coordinated sensorimotor dynamics between the two individuals, with adjustments in velocity, deceleration, and timing of movement that enabled tendrils to meet in a precise region of space. Notably, a complementary division of motor labors, or roles, emerged. One plant displayed a trajectory more oriented toward the partner (a “handler” profile), while the other performed the final approach and grasp (the “grasper”), initiating the physical intertwine through an energy-demanding movement phase. These findings provide compelling evidence for a form of joint action, commonly defined as a coordinated motor strategy to achieve a shared goal, in plants [23] suggesting that cooperation and role specialization can arise in the absence of a nervous system. This work further underscores that complex interactive behaviors, both competitive and cooperative, may rely on fundamental sensorimotor mechanisms embedded in biological systems far removed from neural complexity.

More distributed social behaviors are mediated through mycorrhizal networks—symbiotic associations between plant roots and fungi (Figure 2C). Simard et al. [46] showed that such networks connect species like *Betula papyrifera* and *Pseudotsuga menziesii*, enabling carbon and nutrient exchange. Though these species compete for resources, they also offer mutual support: birches transfer carbon to shaded firs in summer, and firs reciprocate in autumn when birches become carbon-limited, reflecting mutualism based on physiological need and context [46,99].

Further research has identified older individuals with extensive root and mycorrhizal connections, that act as hubs in forest communities [100]. These trees channel resources to younger or stressed kin, facilitating regeneration. Genetic analysis has confirmed preferential support among related trees, suggesting kin recognition and differential resource allocation [100,101].

Even more striking is the case of tree stumps maintained alive by neighboring trees. Bader and Leuzinger [102] observed water and nutrient flow between Kauri trees (*Agathis australis*) and adjacent stumps, likely via root grafts. Though stumps lack photosynthetic tissue and cannot survive alone, they persist by drawing resources from neighbors. Trees may gain structural stability and extended soil access through shared roots. This interdependence suggests forests function less as collections of individuals and more as integrated superorganisms [99,101,103,104]. In drought, such cooperation may buffer environmental stress, with resource-rich trees supporting others.

These findings highlight a hidden, yet vital system often called the “wood-wide web” [105]. This subterranean social network allows plants to share not only resources but also information, coordinating their behavior. The concept of a social network, traditionally applied to humans, is now extendible to plant communities. As Scott [106] noted, social networks illuminate relational structures. Jeong [107] applied sociograms, tools used in human social network analysis, to plant systems, revealing interaction patterns and distributions not detectable through traditional ecological methods. These insights underscore how plant communities self-organize and adapt through sophisticated social interactions.

More than two centuries ago, Alexander von Humboldt described a “socially organized plant life” [108]. Today, modern science is uncovering the mechanisms behind these early observations. As our understanding of cooperation among plants and microbes grows, it becomes essential to develop new frameworks for studying social behavior across taxa, and the complex, cooperative relationships in the ecosystems [101].

Despite the vast diversity of organisms and ecological contexts, cooperative behaviors exhibit several key invariants, fundamental principles that remain consistent across biological systems. These include role differentiation and division of labor, where individuals adopt specialized roles that may incur personal costs but enhance the overall fitness of the group. Another core principle is communication and coordination, which depends on information exchange, ranging from *quorum sensing* in bacteria and Archaea, to chemical signaling in plants, and vocalizations in birds. Additionally, resource sharing and mutual support are common, involving the redistribution of resources in ways that balance individual costs with collective benefits, as seen in microbial cooperation and root-nutrient exchange among plants.

These invariants suggest that cooperation is not merely an adaptive strategy but a fundamental organizing principle of life. Recognizing these patterns provides a conceptual framework to study social behavior and collective dynamics in diverse ecological and evolutionary contexts (see Table 3).

## 5. Conflict Policing

As defined by Miller [109], competition refers to the active demand for a shared and limiting resource by two or more individuals—either within the same species (intraspecific competition) or between different species (interspecific competition). This interaction has long been recognized as a key driver of natural selection, forming a cornerstone of Darwin’s theory of evolution [110]. Conflict interactions structure populations and communities, influence fitness landscapes, and shape trait evolution across taxa.

### 5.1. Conflict Policing in Eusocial Insects

Conflict is an inevitable consequence of social interactions, even in cooperative or kin-structured groups. In Animalia, various mechanisms have evolved to mitigate such conflicts, maintaining collective function while limiting the costs of cheating or overexploitation [111]. In social animals, conflict policing is often enacted by dominant individuals or specialized “policer” members of the group [112]. For example, in eusocial insects like ants, bees (e.g., *Apis Mellifera*), and wasps, worker policing through chemical pheromones prevents subordinate reproduction through aggression, egg destruction, or behavioral inhibition, ensuring that resources are preferentially allocated to the queen’s offspring and stabilizing colony-level cooperation [113].

### 5.2. Conflict Policing in Microbial Domain

In *Bacteria* and *Archea* domains, microorganisms typically coexist in dense, multispecies communities where they must compete for limited nutrients and spatial resources [114]. In response to these challenging conditions, microbes have evolved a wide array of competitive phenotypes. These include the secretion of enzymes to acquire nutrients, the loss of metabolically expensive genes when their products can be obtained from neighboring organisms, direct physical attacks such as piercing or poisoning adjacent cells, and the occupation of space in ways that prevent colonization by competitors [114].

Microbes like *Saccharomyces cerevisiae* (*Eukarya* domain, *Fungi* kingdom) and *Escherichia coli* (*Bacteria* domain) engage in metabolic competition by switching from fermentation to respiration in the presence of oxygen. This shift favors rapid growth at the cost of lower yield, enabling them to uptake nutrients more quickly than their rivals [115]. Other microorganisms, such as *Myxococcus xanthus* (*Bacteria* domain) and *Dictyostelium discoideum* (*Eukarya* domain, *Protists* kingdom), form multicellular fruiting bodies that facilitate movement toward food sources and restrict the diffusion of extracellular digestive enzymes. These structures confer dual benefits: enhanced motility to explore new niches and cooperative adhesion among genetically similar cells, thereby maximizing biomass accumulation and excluding intraspecific competitors [116,117,118].

Another form of antagonism is “interference competition,” wherein organisms directly hinder the performance of others. For example, *Bacillus subtilis* secretes enzymes that degrade QS molecules produced by *Vibrio cholerae*, thereby inhibiting its ability to form biofilms [119,120].

In this sense, it would be interesting to consider whether traits such as quorum sensing (QS), which mediate competitive interactions, are maintained or have analogs in distant taxa. For instance, in plants where competitive behavior occurs through allelopathy (i.e., the release of biochemicals that negatively affect the germination, growth, survival, or reproduction of neighboring plants). As noted above in the “Communication and Signaling” section, this could represent a co-occurrence of similar forms of social behavior arising along different evolutionary trajectories.

### 5.3. Conflict Policing in Plants

For instance, volatiles released by *Artemisia tridentata* have been shown to inhibit seed germination in *Nicotiana attenuate* [121]. Similarly, exposure of lettuce (*Lactuca sativa*), barnyard grass (*Echinochloa crus-galli*), and wheat (*Triticum aestivum cv. Grana*) to volatiles emitted from pulverized leaves of certain *Brassicaceae* species—particularly *Brassica nigra* and *B. juncea*, resulted in delayed germination and reduced growth [122,123]. Allelochemicals can be released into the rhizosphere [124,125], emitted as VOCs [49], or even deposited in reproductive structures like pollen [126]. These compounds vary in their mechanisms of action and uptake, with effectiveness influenced by factors such as concentration, spatial distribution, timing of release, and environmental conditions [127,128,129]. Importantly, allelopathy may be temporally regulated, occurring predominantly during key developmental windows such as seedling establishment, thereby providing an early competitive advantage while minimizing long-term metabolic costs [20,123].

Further interest in the intraspecific competition of plants has been extended to its genetic basis [130,131,132,133]. A seminal study by Dudley and File [29] revealed that *Cakile edentula*, an annual plant, can recognize kin in competitive contexts and modulate its behavior accordingly. Plants grown in proximity to close relatives invested less in root biomass, a proxy for below-ground competitive effort, compared to those growing near unrelated individuals. This kin recognition appears to reduce intra-competition, promoting group-level benefits. While kin recognition and kin selection are well-documented in animals [134,135], their manifestation in plants remains underexplored and conceptually less intuitive [136].

Compelling evidence of dominance-based competitive behavior (Figure 2D), functionally analogous to territoriality in animals, has also been observed in plants. Bonato et al. [22] studied social dynamics in climbing pea plants *(Pisum Sativum)* competing for a single vertical support, a key resource that enables vertical growth, better light access, and survival. Using innovative 3D kinematic analysis—commonly used in behavioral science—they precisely measured circumnutation, the plant’s rhythmic, helical movement that helps locate supports.

Their findings showed one plant usually dominated quickly coiling around the support, while the other slowed and redirected growth to avoid competition. This behavior suggests a competitive decision-making process, similar to animal territorial disputes, where the subordinate plant balances the risk of sharing a support against exploring new territory for potential gain.

The implications of these findings are significant. They suggest that plants are capable of sophisticated sensorimotor interactions with their environment, governed by context-sensitive behavioral modulation. Importantly, the mechanisms behind this modulation likely involve root-mediated communication, potentially through chemical exudates that inform the plant of nearby individuals’ presence, identity, and competitive status [58]. These root-level signals may then influence aboveground growth dynamics, shaping tendril behavior in functionally adaptive ways.

That such behaviors bear a strong resemblance to animal-like territoriality reinforces the idea that fundamental competitive strategies may be deeply rooted in biological systems, emerging from core interactional principles such as self–non-self discrimination and environmental feedback, rather than from higher-order cortical processing. This convergence between plant and animal behavior underscores the value of adopting a cross-taxa perspective in the study of sociality, as it reveals common evolutionary solutions to challenges of resource allocation, spatial negotiation, and inter-individual communication.

Another notable competitive strategy is information theft or “eavesdropping”, which is common in animals [137] and increasingly recognized in plants. In animals, eavesdropping allows individuals to assess rivals or potential allies without direct confrontation [138]. Plants similarly “listen” to chemical signals from their neighbors throughout their lives. Plants use volatile organic compounds (VOCs) to communicate a wide range of ecological information, such as signaling herbivore attack [51,52,53,54], attracting pollinators [54], and mediating environmental stress responses [56]. VOC-mediated signaling plays a crucial role in both above- and below-ground interactions and may be more finely tuned among genetically related individuals [57]. Nonetheless, unrelated plants can exploit these cues to trigger defensive or adaptive responses that enhance their own fitness [58,59,60]. To date, plants eavesdrop on volatiles emitted by herbivore-attacked neighbors to prime their own defenses [139,140,141], and similarly extract information via mycorrhizal networks connecting root systems [142,143]. Parasitic plants also exploit this form of eavesdropping. *Cuscuta pentagona*, an obligate parasite, eavesdrops on host-specific volatiles to locate and grow toward suitable hosts [144].

Across all domains of life, conflict emerges as an inevitable consequence of the active demand for limiting resources, whether nutrients, space, or reproductive opportunities. Despite the immense diversity of mechanisms, several invariant principles appear to unify conflict regulation across taxa. Chemical mediation appears a pervasive feature. In Bacteria and Archaea, conflict and policing are often based on secreted metabolites, toxins, or enzymes that interfere with competitors’ quorum sensing or growth. In Plantae, volatile organic compounds, root exudates, and allelopathic chemicals regulate both competitive exclusion and kin-sensitive modulation of investment, paralleling bacterial strategies of interference and information theft, while in Animalia as social insects, chemical cues such as pheromones underpin dominance interactions and worker policing, stabilizing group-level cooperation. Further, it seems that conflict regulation systems across taxa share an evolutionary logic of balancing competition and cooperation. This convergence highlights how life repeatedly evolves analogous solutions to the universal problem of conflict, underscoring conflict policing as a central and conserved axis of social evolution (see Table 4).

## 6. From Genes to Selection Gradients

What we reported so far, suggests convergent behavioral strategies across phylogenetically distant taxa, claiming that sociality constitutes an evolutionarily conserved adaptation. Recognition systems, cooperative behaviors, conflict policing, and communicative behaviors are observed across all three domains of life—*Bacteria*, *Archaea*, and *Eukarya*—indicating that the emergence of social behavior does not require the emergence of a central nervous system.

This challenges neurocentric frameworks, including the “social brain” hypothesis [3] and classical cognitivist paradigms [145] which privilege vertebrate models and emphasize species-specific neural mechanisms. Sociality appears to be a foundational evolutionary strategy, deeply embedded within molecular and genetic systems.

The genetic code, which governs translation from the four-letter nucleic acid alphabet to the twenty-letter amino acid alphabet, occupies a central role in biology [146,147]. It is arguably the most universal informational system across all life forms. Despite some variations identified through studies of diverse organisms, the basic structure and most codon assignments are strikingly conserved [148,149].

In this light, sociality with both its competitive and cooperative modes may represent one of the earliest forms of adaptation, potentially originating in the Last Universal Common Ancestor (LUCA), the hypothetical progenitor of all current life [150]. The “Genetic toolkit hypothesis” [151,152], based on evo-devo principles, posits that regulatory changes in conserved genes, pathways, or networks drive the evolution of novel phenotypes. The involvement of genes in social behaviors across taxa is supported by transcriptomic research in simple organisms such as microbes [86,88,153]. In some microbes, a distinct social stage exists, and the genes regulating this shift can be identified by comparing gene expression across life cycle stages. Transcriptomic analyses have defined cooperative genes as those highly expressed during social stages but not during solitary ones [154]. A similar methodology has been successfully applied in social insects too [155,156].

Together, these gene-based mechanisms support the hypothesis that the capacity for social interaction is encoded in the genome, enabling the synthesis of proteins and receptors essential for self and other’s recognition [157]. In particular, specific comparative genomics studies have highlighted orthologous gene families activated in diverse lineages and species for social behaviors [158]. By comparing brain transcriptomes from honey bees, mice, and sticklebacks fishes, Seul and colleagues [158] identified orthologous gene families and co-expressed modules associated with social challenge responses, revealing a conserved neural ‘toolkit’ for social behavior across evolutionarily distant species.

Classical social evolution models, such as kin selection, multilevel selection, and ecological constraints, provide a foundational framework for understanding the evolutionary pressures shaping social behaviors. Kin selection explains how genes promoting altruistic behaviors can be favored if they increase inclusive fitness, that is, the reproductive success of genetically related individuals. Multilevel selection considers the simultaneous action of selection at both the individual and group levels, favoring traits that enhance group cohesion and cooperation for collective benefit. Ecological constraints models emphasize how environmental conditions influence the expression of social traits.

The concept of a genetic toolkit integrates these models by highlighting the shared and conserved molecular substrate, composed of gene families and regulatory networks that control the development and expression of social behaviors. These toolkits provide the necessary flexibility for selective pressures to act by modulating gene activation in response to different genetic, social, and ecological contexts. Thus, selection processes explain why certain social behaviors are favored, while genetic toolkits explain how these behaviors are manifested at the molecular level, bridging ultimate and proximate explanations in social biology.

In this light, current evidence suggests that social behavior may often emerge from conserved molecular and genetic architectures, and is not a property reserved for highly neuronal minds; yet it is not dictated solely by the genome, as environmental cues in socially polymorphic species [159] can modulate, override, or reshape these predispositions, revealing sociality as a dynamic interplay between genes and context rather than a fixed, genomic script (e.g., ants [160]).

Recognizing social behavior as influenced by genetic factors encourages us to reconsider the boundaries of individuality, agency, competition, and cooperation across the tree of life. As modern biological tools continue to reveal the underlying architectures of interaction encoded in the genome, it becomes clear that sociality is not only a consequence of evolution, but may also play a significant role in shaping it.

## 7. Conclusions

The evidence reviewed here underscores that sociality is a pervasive and evolutionarily conserved feature of life, extending far beyond organisms with complex nervous systems. Across bacteria, archaea, fungi, plants, and animals, common molecular and genetic mechanisms underpin recognition, communication, cooperation, and conflict regulation. These patterns suggest that social behaviors are scaffolded by fundamental biological architectures, deeply embedded in the genome and modulated by environmental and social contexts.

In our opinion, plants in particular provide a compelling case for expanding our understanding of sociality. Despite lacking a nervous system, they demonstrate kin recognition, chemical and electrical signaling, growth modulation in response to neighbors, and coordinated defense strategies [101]. Integrating plant models with microbial and animal systems highlights the possibility of a minimal molecular “social toolkit,” rooted in conserved genes and regulatory networks, yet flexible enough to adapt to ecological and evolutionary pressures.

This perspective emphasizes sociality not as a consequence of cognitive complexity but as a fundamental evolutionary strategy, potentially present since the Last Universal Common Ancestor (LUCA). The interplay between conserved genomic toolkits and environmental cues reveals sociality as a dynamic, emergent property rather than a fixed behavioral script [151,152].

Future research should focus on mapping these conserved molecular circuits across phylogenetically distant taxa, experimentally testing the regulatory and functional principles of social behaviors, and integrating ecological, genetic, and transcriptomic data. Such an approach promises to unify ultimate and proximate explanations of social evolution and may reveal sociality itself as a driving force shaping life’s diversity.

Ultimately, recognizing sociality as a genomic and ecological imperative compels us to rethink classical boundaries of individuality, cooperation, and competition, offering a truly integrative framework for the comparative biology of social behavior.

## Figures and Tables

**Figure 1 biology-14-01239-f001:**
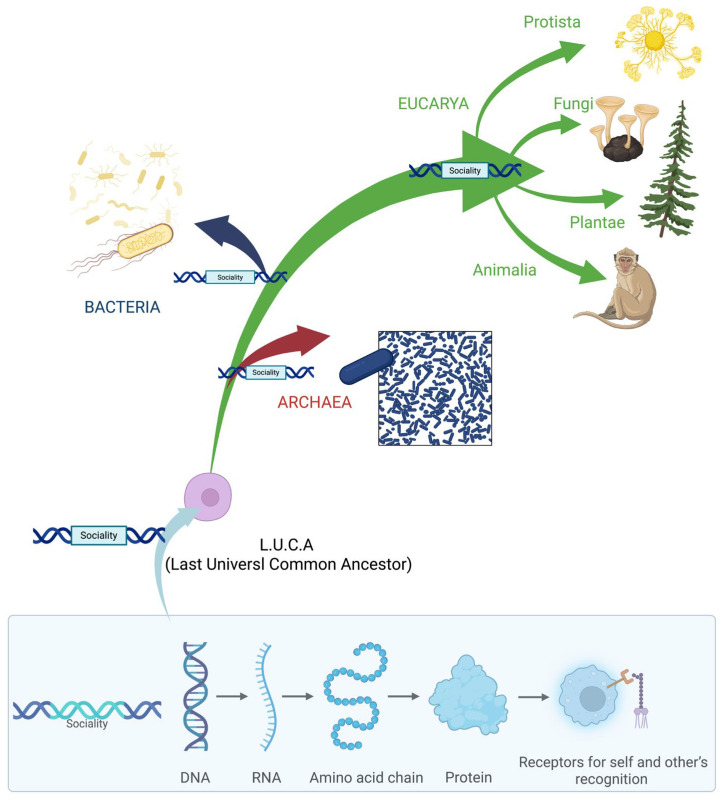
A graphical illustration showing how different domains present social contexts that require the development of social behaviors. The common foundation across these domains is not a centralized brain or nervous system, but rather the DNA code and genetic toolkit that enable the encoding of specific abilities. These abilities enhance survival and can be expressed when organisms encounter suitable environmental conditions.

**Figure 2 biology-14-01239-f002:**
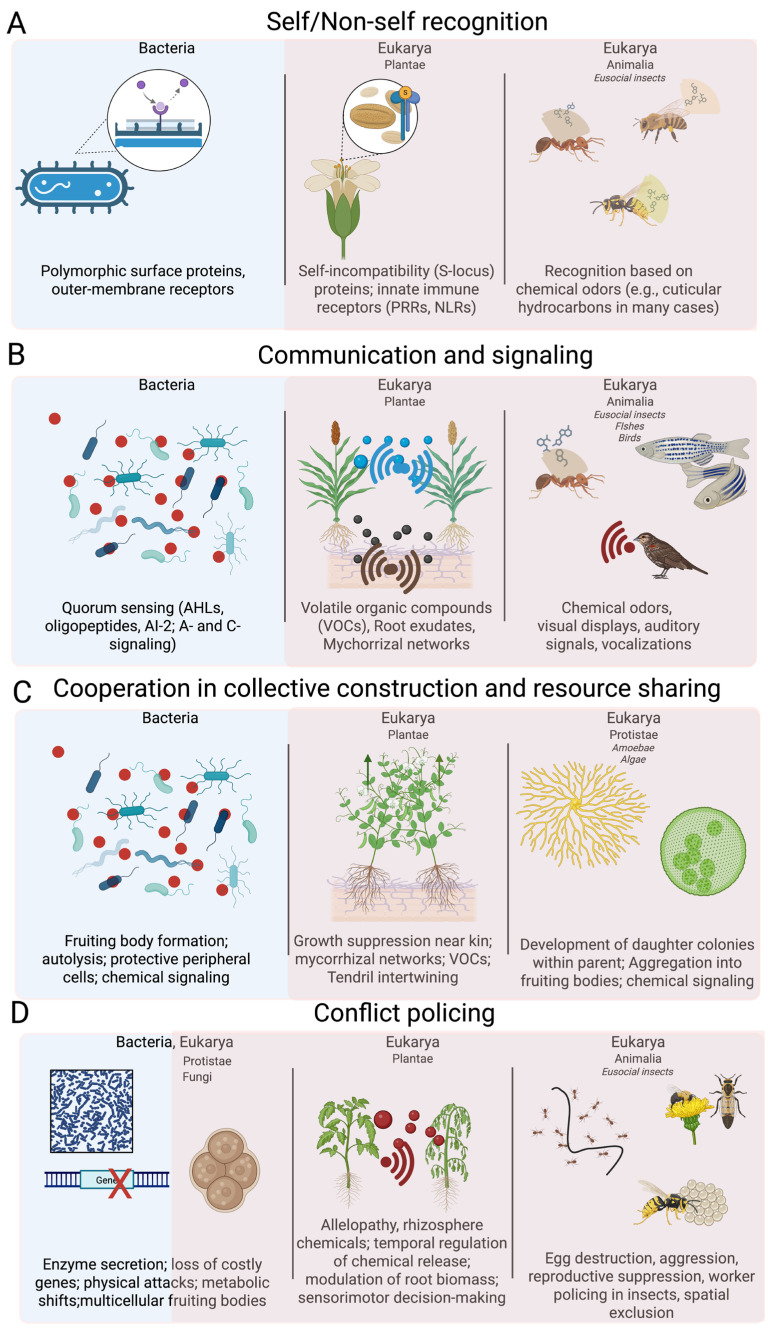
Graphical representation of examples for the main social behaviors summarized in the text. Panel (**A**) reports examples of self/non-self-recognition across taxa; in panel (**B**) are reported. Examples of communication and signaling across taxa. Panel (**C**) reports examples of cooperative behaviors across taxa. Panel (**D**) reports examples of conflict policing across taxa. Blue background refers to bacteria domain, while soft red background refers to Eukarya domain (Plantae, Protists, Fungi and Animalia kingdoms).

**Table 1 biology-14-01239-t001:** Summary table for self/non-self-recognition in different domains reported in the paragraph.

Domain/ Examples	Mechanism/ Molecular Basis	Role in Social Behavior	Reference
Bacteria (*Proteus mirabilis*, *Bacillus subtilis*)	Polymorphic surface proteins, outer-membrane receptors	Kin recognition, territoriality, conflict resolution, restrict cooperation	Tsutsui, 2004 [25]; Breed & Bennett, 1987 [26]
Eukarya Plantae (*Arabidopsis thaliana*, *Cakile Edentula*, *B. rapa*, *B. olerosa*)	Self-incompatibility (S-locus) proteins; innate immune receptors (PRRs, NLRs)	Prevent self-fertilization, promote outcrossing; discriminate self vs. non-self roots; pathogen recognition	Takayama & Isogai, 2005 [28]; Dudley & File, 2007 [29]; Jones & Dangl, 2006 [30]
Eukarya Animalia (*Apis mellifera*, *Camponotus vagus*, *Camponotus floridanus*, *Lassioglossum zephyrum*, *Polistes* spp., *Vespula* spp.)	Recognition based on chemical odors (e.g., cuticular hydrocarbons in many cases)	Colony recognition, nestmate discrimination, caste/task signaling	Breed & Bennett, 1987 [26]; Gamboa & Fred, 2004 [34]; Tsutsui, 2004 [25]

**Table 2 biology-14-01239-t002:** Summary table for communication and signaling in different domains reported in the paragraph.

Domain/ Examples	Mechanism/ Molecular Basis	Role in Social Behavior	Reference
Bacteria (*Pseudomonas* *aeruginosa*, *Myxococcus xanthus*)	Chemical signaling: quorum sensing (AHLs, oligopeptides, AI-2; A- and C- signaling in M. xanthus)	Kin recognition, cooperation, conflict resolution, aggregation, fruiting body formation, spore differentiation	Keller & Surette, 2006 [36]; Griffin et al., 2004 [39]; Fiegna & Velicer, 2005 [40]; Kaiser, 2004 [42]; Shimkets, 1999 [44]; Strassman, 2000 [45]
Eukarya Plantae (*Phaseolus vulgaris*, *Zea Mays*, *Nicotiana tabacum*, *Oryza sativa*, *Gossypium hirstum*, *Pisum sativum*, *Artemisia Tridentata*)	Volatile organic compounds (VOCs), Root exudates.	Herbivore defense, pollinator attraction, neighbor communication, pathogen defense, adaptation to stress	Dicke & Sabelis, 1988 [51]; Turlings et al., 1990 [52]; De Moraes et al., 2001 [53]; Dudareva & Pichersky, 2000 [55]; Karban et al., 2006 [57]; Gagliano & Grimonprez, 2015 [48]; Karban & Shiojiri, 2008 [49].
Eukarya Animalia Insects (*Camponotus vagus*, *Camponotus floridanus*, *Linepithema humile*, *Solenopsis invicta*, *Apis mellifera*, *Lasioglossum zephyrum*, *Polistes* spp., *Vespula* spp.) Fish (*Ostariophysi*, *Cichlidae*, *Gobidae*, *Pomacentridae*, *Porichthys notatu*) Birds (*Parus minor*, *Troglodytes aedon*, *Taeniopygia guttata*, *Parus* spp., *Junco hyemalis*) Fungi (*Saccharomyces cerevisiae*)	Chemical odors, visual displays, auditory signals, vocalizations	Colony recognition, nestmate discrimination, social hierarchy maintenance, territoriality, mate choice	Breed & Bennett, 1987 [26]; Gamboa, 2004 [34]; Tsutsui, 2004 [25]; Ladich, 2019 [76]; Bass & McKibben, 2003 [77]; Suzuki, 2016 [78]; Kroodsma & Miller, 2020 [79]; Grieves et al., 2022 [80]

**Table 3 biology-14-01239-t003:** Summary table for cooperation as collective construction and resource sharing in different domains reported in the paragraph.

Domain/ Examples	Mechanism/ Molecular Basis	Role in Social Behavior	Reference
Bacteria (*Myxococcus xanthus*) Archea	Fruiting body formation; autolysis; protective peripheral cells; chemical signaling	Division of labor among spores, dying, and protective cells; cooperation enables survival under nutrient scarcity; social cheating can evolve	Velicer & Vos, 2009 [89]
Eukarya Plantae (*Pisum Sativum*, *Betula papyrifera*, *Pseudotsuga menziesii*, *Agathis australis*)	Growth suppression near kin; mycorrhizal networks; VOCs; Tendril intertwining	Enhances collective fitness; resource and information sharing; hub trees support younger or stressed trees; “wood-wide web”; Joint action and mutual mechanical support; division of motor roles (handler vs. grasper)	Semchenko et al., 2010 [95]; Dudley, 2015 [97]; Wang et al. [6], 2020; Rowe & Speck, 2015 [99]; Castiello, 2021 [102]; Simard et al., 2009 [100]; Beiler et al., 2010 [101]; Bader & Leuzinger, 2019 [103]; van Der Hejiden et al., 2015 [104]; Jeong, 2024 [108]; Bonato et al., 2024 [23]
Eukarya Protistae (*Dictyostelium discoideum*, *Volvox carteri*)	Development of daughter colonies within parent; Aggregation into fruiting bodies; chemical signaling	Stalk cells sacrifice themselves to elevate spores; enables collective reproduction; cooperative division of labor even among unrelated individuals, Early forms of parental investment; kin-directed cooperation	Kocher, 1968 [98]; Castiello, 2021 [102]; Velicer & Vos, 2009 [89]; Strassman & Queller, 2011 [87]; Bonner, 2008 [86]; Fortunato et al., 2003 [88].

**Table 4 biology-14-01239-t004:** Summary table for conflict policing in different domains reported in the paragraph.

Domain/ Examples	Mechanism/ Molecular Basis	Role in Social Behavior	Reference
Bacteria (*Escherichia coli*, *Myxococcus xanthus*, *Bacillus subtilis*, *Vibrio cholerae*) Protistae (*Dictyostelium discoideum*) Fungi (*Saccharomyces cerevisiae*)	Enzyme secretion; loss of costly genes; physical attacks; metabolic shifts; multicellular fruiting bodies	Limits competitor growth; maximizes resource acquisition; enhances survival under dense community conditions, Interference competition; inhibits biofilm formation of competitors	Ghoul & Mitri, 2016 [115]; Pfeiffer et al., 2001 [116]; Hibbing et al., 2010 [117]; Nadell & Bassler, 2011 [118]; Thiery & Kaimer, 2020 [119].
Eukarya Plantae (*Artemisia tridentata*, *Nicotiana attenuate*, *Lactuva sativa*, *Cakile edentula*, *Echinochloa crus-galli*, *Triticum Aestivum cv. Grana*, *Brassica nigra*, *Brassica juncea*, *Cuscuta pentagona*)	Allelopathy via VOCs, rhizosphere chemicals; temporal regulation of chemical release; modulation of root biomass; sensorimotor decision-making	Inhibits germination and growth of neighbors; reduces metabolic cost while gaining early competitive advantage, Reduces intraspecific competition; promotes group-level fitness, Establishes dominance; competitive resource acquisition; resembles territoriality, Assesses competitor presence; primes defenses; adjusts growth; exploits neighbor information	Oleszek, 1987 [123]; Effah et al., 2019 [124]; Kato-Noguchi et al., 2010 [125]; Kong et al., 2018 [126]; Karban et al., 2013 [57]; Falik et al., 2006 [131]; Boyden et al., 2008 [132]; Crustinger et al., 2008 [133]; West et al., 2002 [135]; Tibbetts & Dale, 2007 [136]; Callaway, 2007 [137]; Wang et al., 2021 [96]; Bonato et al., 2025 [72]; Bonato et al., 2023 [22]; Dicke & Sabelis, 1988 [51]; Oliveira & Pie, 1998 [139]; Turlings & Tumlinson, 1990 [52]; Arimura et al., 2000 [140]; Dolch et al., 2000 [141].
Eukarya Animalia (*Formicidae*, *Vespidae*, *Apis Mellifera*)	Egg destruction, aggression, reproductive suppression, worker policing in insects, spatial exclusion	Mantain social cohesion and cooperation, suppresses cheating, stabilizes group-level reproductive investment.	Ratnieks, 1988 [114]; Beekman, 2013 [113]

## Data Availability

Not applicable.

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
