# Peer review of "Key Invariants in the Evolution of Sociality Across Taxa"

_biology, 2025, doi:10.3390/biology14091239_

Round 1
Reviewer 1 Report
Comments and Suggestions for Authors
The review "The evolution of social modes across taxa: a matter of genes" explores the emergence of social traits across different types of life, including bacteria, single-celled organisms, animals, and plants.
The main take-home message is that sociality is not a late-emerging property but an ancient, genome-level adaptation. Across Bacteria, Archaea, and Eukarya, the same conserved genetic pathways for self/non-self-recognition, signalling, and coordinated behaviour give rise to both competition and cooperation. These shared molecular ‘social toolkits’ show that very different organisms converge on similar social solutions, proving that complex nervous systems are optional and not prerequisites for social life.
I think this is a good contribution to the field and of interest to the journal readers. By spanning all three domains of life and asking whether the same genetic circuits underpin both competition and cooperation, it offers a unifying perspective.
That said, there are a couple of areas the review could tighten definitions, reorganize around mechanistic modules, ground broad claims in comparative data, and explicitly highlight alternative explanations.
- The authors treat any cross-taxon similarity as ‘conserved,’ yet examples like quorum sensing vs. plant volatile signalling are likely independent co-options which over-extends ‘deep homology’. This (over)simplification weakens the evolutionary inference. They could add highlights of explicit distinctions: shared ancestry, parallel co-option of the same pathway, and true convergence. Cite some phylogenomic meta-analyses where possible.
- Somehow, the definition and usage of ‘sociality’ is unbounded; the usage ranges from fetal motor coordination in humans to allelopathy in plants without a criterion that would exclude, say, any chemically mediated interaction. Up-front, define sociality then consistently use it within that definition's boundaries unless otherwise stated.
- The review states that sociality ‘may originate in LUCA’ and invokes a ‘genetic toolkit hypotheses,’ but provides no comparative genomics summary. The authors could add a synthesis figure/table summarising orthologous gene family, evidence of selection on social vs. solitary phases, and lineage coverage.
- The argument for a genome-level imperative does not situate gene-level processes within kin selection, multilevel selection, or ecological constraints. They could further integrate ultimate fitness and proximate molecular explanations; reserve a subsection ‘From genes to selection gradients’ linking toolkit genes to classic social-evolution models.
- Instead of the current text layout, which adopts a binary ‘competitive vs. cooperative modes’ sections, re-section by functional modules: 1) Self/Non-self recognition, 2) Communication & signalling, 3) Collective construction & resource sharing, 4) Conflict policing. Each module can then compare Bacteria, Archaea, and Eukarya, ending with a ‘Key invariants’ synthesis.
- The prose blocks are rather lengthy, which reduces readability. The authors could add visual summaries by adding a ‘Road-map’ paragraph, previewing section logic, and creating at least one summary table per module with domain, mechanism, evidence strength, etc., and refs columns.
- To ensure methodological transparency and reproducibility, there should also be a paragraph or section on ‘Literature‐search criteria & inclusion thresholds’ stating the databases searched, date range, keywords, and cut-offs for inclusion.
- Sentences like “sociality is encoded in the genome… reiterated within the genome itself” overstate causality. Replace with conditional phrasing ‘current evidence suggests…’, and juxtapose counter-cases, e.g., socially polymorphic species where environment overrides toolkit genes.
Author Response
1. The authors treat any cross-taxon similarity as ‘conserved,’ yet examples like quorum sensing vs.
plant volatile signalling are likely independent co-options which over-extends ‘deep homology’. This
(over)simplification weakens the evolutionary inference. They could add highlights of explicit
distinctions: shared ancestry, parallel co-option of the same pathway, and true convergence. Cite
some phylogenomic meta-analyses where possible.
R1. We thank the reviewer for this valuable comment. To our knowledge, no studies have linked bacterial
quorum sensing to plant volatile organic compounds (VOCs) through a reconstructed evolutionary or
phylogenetic pathway, and there are no current insights supporting a possible parallel co-option between
these traits in bacteria and plants.
The novelty of this review lies precisely in addressing VOCs in plants, highlighting their potential similarity to
bacterial quorum sensing as a form of chemical competition. While this represents a first step, we emphasize
that we are not claiming a shared evolutionary origin, but it’s more a proposal to discuss. To improve the
manuscript, we have followed the reviewer’s suggestion by clarifying that we use quorum sensing in bacteria
and communication (and specifically also allelopathy) in plants as examples of analogous competitive
strategies, rather than implying direct evolutionary linkage. We also cite DiFrisco et al. (2023) to underscore
the importance of discriminating between co-option, convergence, and shared ancestry. Please refer to pp.9
and 16.
2. Somehow, the definition and usage of ‘sociality’ is unbounded; the usage ranges from fetal motor
coordination in humans to allelopathy in plants without a criterion that would exclude, say, any
chemically mediated interaction. Up-front, define sociality then consistently use it within that
definition's boundaries unless otherwise stated.
R2. We thank the reviewer for this comment. We have revised the manuscript to provide a clear, upfront
definition of “sociality” (please refer to p. 2) and have ensured that the term is now used consistently within
these defined boundaries throughout the text.
3. The review states that sociality ‘may originate in LUCA’ and invokes a ‘genetic toolkit hypotheses,’
but provides no comparative genomics summary. The authors could add a synthesis figure/table
summarising orthologous gene family, evidence of selection on social vs. solitary phases, and
lineage coverage.
R3. We sincerely thank the reviewer for this thoughtful suggestion. We fully agree that a comparative
genomics summary would provide valuable context. However, as our expertise does not lie in genomics per
se, we do not feel confident in providing a comprehensive synthesis table or figure without risking
oversimplification or omission. Instead, we have expanded the relevant section of the manuscript to
incorporate additional references from comparative genomics studies that identify orthologous gene families
associated with social behavior across diverse taxa, from bacteria to mice. These studies support the
hypothesis of a conserved genetic toolkit underlying sociality and reinforce our statement that such
mechanisms may have originated in LUCA. Please refer to p. 20.
4. The argument for a genome-level imperative does not situate gene-level processes within kin
selection, multilevel selection, or ecological constraints. They could further integrate ultimate fitness
and proximate molecular explanations; reserve a subsection ‘From genes to selection gradients’
linking toolkit genes to classic social-evolution models.
R4. Thank you for this suggestion. We have revised the final section discussing genes, renaming as “From
genes to selection gradients”, and have integrated references to some classic social-evolution models. This
should better situates gene-level processes within kin selection, multilevel selection, and ecological
constraints, linking ultimate fitness with proximate molecular explanations. Please refer to p.20-21.
5. Instead of the current text layout, which adopts a binary ‘competitive vs. cooperative modes’
sections, re-section by functional modules: 1) Self/Non-self recognition, 2) Communication &
signalling, 3) Collective construction & resource sharing, 4) Conflict policing. Each module can then
compare Bacteria, Archaea, and Eukarya, ending with a ‘Key invariants’ synthesis.
R5. We thank the reviewer for this suggestion. We have now restructured the text accordingly, moving away
from the binary “competitive vs. cooperative modes” layout. The manuscript is now organized into four
functional modules: (1) Self/Non-self recognition, (2) Communication & signalling, (3) Collective construction
& resource sharing, and (4) Conflict policing. Each section compares mechanisms across Bacteria, Archaea,
and Eukarya, and I conclude with a dedicated key invariant synthesis that highlights convergent principles
across domains. We have also created a new figure 2 that encompassess all the examples described in the
main text for each paragraph.
6. The prose blocks are rather lengthy, which reduces readability. The authors could add visual
summaries by adding a ‘Road-map’ paragraph, previewing section logic, and creating at least one
summary table per module with domain, mechanism, evidence strength, etc., and refs columns.
R6. Thank you for this. To improve readability and provide visual summaries, we have now included
a summary table at the end of each section. Each table reports the relevant domain, mechanism, role in
social behavior, and supporting references, thereby offering a concise overview that complements the longer
prose.
7. To ensure methodological transparency and reproducibility, there should also be a paragraph or
section on ‘Literature-search criteria & inclusion thresholds’ stating the databases searched, date
range, keywords, and cut-offs for inclusion.
R7. By following the reviewer’s suggestion we have added a new section entitled “Literature-search criteria &
inclusion thresholds” in Appendix I, clearly reporting the databases searched, date range, keywords, and
inclusion/exclusion criteria to ensure methodological transparency and reproducibility. We placed this section
in the Appendix in order to keep the main text concise and focused, while still allowing interested readers to
access the complete methodological details.
8. Sentences like “sociality is encoded in the genome… reiterated within the genome itself” overstate
causality. Replace with conditional phrasing ‘current evidence suggests…’, and juxtapose countercases,
e.g., socially polymorphic species where environment overrides toolkit genes.
R8. We have revised the sentence to adopt conditional phrasing, emphasizing that current
evidence suggests a genetic and molecular basis for social behavior, without implying strict causality. We
also highlighted counter-examples, such as socially polymorphic species where environmental factors can
modulate or override genetic toolkits, and explicitly noted that sociality is not a property reserved for highly
neuronal minds. Please refer to p. 21.
Reviewer 2 Report
Comments and Suggestions for Authors
The review manuscript titled “The evolution of social modes across taxa: a matter of genes” was an intriguing read and the point of considering social behaviour in a context without complex neural networks is a fresh idea that is potentially relevant to a wide audience. The manuscript is nicely written and combines new knowledge with historical ideas pointing to a potentially new framework of thinking about sociality. However, as is, this review seems to collate a somewhat selected sample of cases from different groups of organisms, but does not attempt a meta-analysis or a comprehensive review across taxa. Is the review suggesting a new idea based on a comprehensive literature review trying to solve an important question, or collating evidence for a pre-existing idea more like an opinion piece?
Another critical comment has to do with a minor discrepancy between the framing of the work and the actual content. The short summary, abstract and introduction promise a lot for the review, but after the two sections and short conclusions I was left expecting more processing of the findings and building the framework for future studies. I also urge the writes to continue thinking about the new framework these findings point us to, and suggest adding that conceptually to Figure 1, as this could increase the value and benefit of this work to the scientific community.
Similarly, the review provides no new direct evidence of genes involved in sociality, although the title infers such a connection. Therefore, before publishing I suggest formulating the title as a question (by adding a question mark in the end) or changing the title altogether, into something more on the lines of “Examples of social modes across taxa indicate a common origin for sociality”.
Please find more comments and questions below.
Figure 1. The figure legend has an extra “ in the end
Line 215 has an extra comma at the end of the first sentence in the paragraph.
L214-256 This example has been explained in much more detail than the other examples in this review, and I am not sure all the information given is needed for this manuscript. Please revise and shorten.
L252-253 Where do the authors get the idea that the modulation described requires root-mediated communication? If the plant that reaches the top before the other shadows it, isn’t the other plant simply growing to another direction to get light?
L312-317 In the example of Protists, it is explained that genetically unrelated individual amoebas can form mixed groups in which some cells do not reproduce but form a stalk for the reproducing cells with the cost of cell death. How do we know that this is a form of co-operation and not exploitation of weaker amoebas by stronger amoeba cells, i.e. competition? Or on the other hand, is it known whether the cells that adopt the non-reproductive role are actually capable of becoming reproductive, or if they are already at the end of their life-cycle or unable to allocate resources to reproduction?
L400-402 The ending of the discussion refers to another paper by one of the authors calling for new frameworks for studying social behaviour. Could the authors add more discussion about the new framework and its implications?
Author Response
1. Is the review suggesting a new idea based on a comprehensive literature review trying to solve an
important question, or collating evidence for a pre-existing idea more like an opinion piece?
R1. Our review is primarily based on ideas and evidence already proposed in the literature, but we have
extended the discussion to include plants, highlighting their volatile-mediated interactions. This represents a
novel contribution, providing a first step toward exploring how sociality might be conceptualized across highly
divergent taxa and identifying potential common grounds for chemical-mediated interactions between
microbes and plants.
2. Another critical comment has to do with a minor discrepancy between the framing of the work and
the actual content. The short summary, abstract and introduction promise a lot for the review, but
after the two sections and short conclusions I was left expecting more processing of the findings and
building the framework for future studies. I also urge the writes to continue thinking about the new
framework these findings point us to, and suggest adding that conceptually to Figure 1, as this could
increase the value and benefit of this work to the scientific community.
R2. With the revisions we have made throughout the manuscript, we hope to have addressed these
concerns more comprehensively. In particular, we have clarified our intention to provide a broader
perspective that explicitly includes plants, highlighting their relevance to the study of sociality. More
generally, we aim to offer an integrated overview, emphasizing conserved molecular, genetic, and behavioral
principles across diverse taxa. In revising the manuscript, we have also expanded the discussion to include
aspects of self–non-self recognition and communication, in addition to the themes of cooperation and
competition previously addressed. Finally, we have extended the concluding section to articulate our
perspective on the importance of including plants in the study of sociality, and to argue for a conceptual shift:
from viewing sociality as a phenomenon tied exclusively to the presence of a brain, to recognizing it more
fundamentally as a matter of genes and molecular interactions. We believe these additions make our
perspective clearer, more comprehensive, and more valuable to the scientific community.
3. Similarly, the review provides no new direct evidence of genes involved in sociality, although the title
infers such a connection. Therefore, before publishing I suggest formulating the title as a question
(by adding a question mark in the end) or changing the title altogether, into something more on the
lines of “Examples of social modes across taxa indicate a common origin for sociality”.
R3. We thank the reviewer, and we have now changed the title as a question: “Is the evolution of sociality
across taxa a matter of genes?”.
4. Figure 1. The figure legend has an extra “ in the end
R4. We have now removed it, thank you.
5. Line 215 has an extra comma at the end of the first sentence in the paragraph.
R5. Adjusted, thank you.
6. L214-256 This example has been explained in much more detail than the other examples in this
review, and I am not sure all the information given is needed for this manuscript. Please revise and
shorten.
R6. We have now shorten and revised the portion of text. Please refer to 16.
7. L252-253 Where do the authors get the idea that the modulation described requires root-mediated
communication? If the plant that reaches the top before the other shadows it, isn’t the other plant
simply growing to another direction to get light?
R7. The idea that modulation of growth behavior requires root-mediated communication is grounded in
extensive evidence showing that plants detect neighbors through chemical signals, such as root exudates
and volatile organic compounds. These cues allow plants to sense the presence of nearby individuals and
assess their environment accordingly.
In this study’s context, plants perceive their neighbors through both root systems and airborne chemicals,
which informs their growth decisions. While it is true that a plant growing later and facing shadowing may
redirect growth to access light, this response is not simply a passive reaction but a socially mediated
behavior. Plants behave differently when alone compared to when in the presence of neighbors, as
demonstrated by Bonato et al. (2023, 2024), Dudley & File (2007), Karban et al., (2014), Crepy & Casal
(2015) and others. The detection of neighbors triggers adaptive movement patterns that reflect competition
and social context, rather than just individual responses to light availability.
8. L312-317 In the example of Protists, it is explained that genetically unrelated individual amoebas can
form mixed groups in which some cells do not reproduce but form a stalk for the reproducing cells
with the cost of cell death. How do we know that this is a form of co-operation and not exploitation of
weaker amoebas by stronger amoeba cells, i.e. competition? Or on the other hand, is it known
whether the cells that adopt the non-reproductive role are actually capable of becoming reproductive,
or if they are already at the end of their life-cycle or unable to allocate resources to reproduction?
R8. We thank the reviewer for raising this important point. We have now revised the text to clarify this part.
Please refer to p. 12. To better explain, several studies have shown that the cells which differentiate into the
sterile stalk are indeed capable of becoming reproductive spores under other conditions, indicating that their
fate is not predetermined by senescence or resource depletion but involves an active developmental
decision (Strassmann & Queller, 2011). This is precisely why the system has become a model for studying
social evolution and conflict: mixed chimeric groups can harbor “cheater” genotypes that preferentially
contribute to the spore pool while exploiting unrelated cells that form the stalk (Strassmann et al., 2000).
Thus, stalk formation is generally interpreted as a cooperative trait because it enhances dispersal and
survival of the group, but it is simultaneously vulnerable to exploitation, which has driven the evolution of
policing and kin-discrimination mechanisms (Fortunato et al., 2003).
In summary, stalk differentiation is not simply the fate of weak or senescent cells, but a socially relevant
decision that can be cooperative in clonal groups while manifesting as competitive exploitation in chimeric
groups.
9. L400-402 The ending of the discussion refers to another paper by one of the authors calling for new
frameworks for studying social behaviour. Could the authors add more discussion about the new
framework and its implications?
R9. We have now added a conclusion section in which we re-take our opinion on the importance of including
plants in the discussion presented in the review. Please refer to p.21-22
Round 2
Reviewer 1 Report
Comments and Suggestions for Authors
None.
Author Response
Thank you for your revisions.